# Integrative Proteomic and Phosphoproteomic Analyses Revealed Complex Mechanisms Underlying Reproductive Diapause in *Bombus terrestris* Queens

**DOI:** 10.3390/insects13100862

**Published:** 2022-09-23

**Authors:** Yan Liu, Ruijuan Wang, Long Su, Shan Zhao, Xiaoyan Dai, Hao Chen, Guang’an Wu, Hao Zhou, Li Zheng, Yifan Zhai

**Affiliations:** 1Institute of Plant Protection, Shandong Academy of Agricultural Sciences, 23788 Gongye North Road, Jinan 250100, China; 2Key Laboratory of Natural Enemies Insects, Ministry of Agriculture and Rural Affairs, Jinan 250100, China; 3Shandong Provincial Engineering Technology Research Center on Biocontrol of Crops Pests, Jinan 250100, China

**Keywords:** *Bombus terrestris*, diapause, TMT quantitative proteomics, TMT quantitative phosphoproteomics

## Abstract

**Simple Summary:**

*Bombus terrestris* is one of the most ideal pollinators in the world and brings great economic benefits through the pollination of fruits and vegetables. The huge demand for pollinators, combined with habitat loss, climate change, and pesticide use over the past decades, has resulted in a dramatic decline in wild bumblebee populations and distribution. Hence, the artificial breeding of bumblebees is of great significance and has a broad market prospect. Diapause is an important process in artificial breeding. Although some studies have been carried out on the diapause-related genes of bumblebees, the precise mechanisms affecting the phosphorylation level remain unclear. Here, we performed a new comparison of three diapause-stage expression profiles using isobaric tandem mass tag (TMT)-labeled proteomics and phosphoproteomics. The results provided abundant resources and contributed to a better understanding of the mechanisms underlying the regulation of reproductive diapause in eusocial insects.

**Abstract:**

Reproductive diapause is an overwintering strategy for *Bombus terrestris*, which is an important pollinator for agricultural production. However, the precise mechanisms underlying reproductive diapause in bumblebees remain largely unclear. Here, a combination analysis of proteomics and phosphoproteomics was used to reveal the mechanisms that occur during and after diapause in three different phases: diapause (D), postdiapause (PD), and founder postdiapause (FPD). In total, 4655 proteins and 10,600 phosphorylation sites of 3339 proteins were identified. Diapause termination and reactivation from D to the PD stage were characterized by the upregulation of proteins associated with ribosome assembly and biogenesis, transcription, and translation regulation in combination with the upregulation of phosphoproteins related to neural signal transmission, hormone biosynthesis and secretion, and energy-related metabolism. Moreover, the reproductive program was fully activated from PD to the FPD stage, as indicated by the upregulation of proteins related to fat digestion and absorption, the biosynthesis of unsaturated fatty acids, fatty acid elongation, protein processing in the endoplasmic reticulum, and the upregulation of energy-related metabolism at the phosphoproteome level. We also predicted a kinase–substrate interaction network and constructed protein–protein networks of proteomic and phosphoproteomic data. These results will help to elucidate the mechanisms underlying the regulation of diapause in *B. terrestris* for year-round mass breeding.

## 1. Introduction

Many insect species enter a period of developmental or reproductive stasis, called diapause, as an adaptive strategy to enhance survival in response to unfavorable conditions and seasonal changes [1]. Diapause is a complicated and dynamic developmental process characterized by decreased metabolic activity and increased stress resistance [2]. Various neuroendocrine, molecular, cellular, enzymatic, metabolic, endocrine, and behavioral changes occur during diapause [1,2,3]. Diapause usually occurs at species-specific developmental stages (i.e., embryo, larva/nymph, pupa, and adult) [1,4,5,6]. Reproductive diapause often occurs in overwintering adults upon the arrest of vitellogenesis and ovulation [7]. Diapause is classified into three major successive stages: prediapause, diapause, and postdiapause. Prediapause is defined as the induction and preparation for diapause, while diapause includes the initiation, maintenance, and termination of diapause, and postdiapause is characterized by quiescence and rapid development [1]. During prediapause, insects perceive changes in the environment and regulate multiple physiological processes. Resources are accumulated, and maturation is delayed. During diapause, insects are in a state of “suspended animation”, characterized by a low metabolic rate and enhanced stress resistance. In the postdiapause stage, the organisms return to an active state to complete the life cycle. The biological and cellular processes involved in these major physiological transitions are complex and interesting, especially the transition from diapause to the postdiapause stage, which requires the restarting of different groups of biological processes and needs to be further studied [8,9]. Here, the proteins and phosphoproteins involved in the transitions between the successive stages of diapause in bumblebee (*Bombus terrestris*) queens were investigated.

*B. terrestris* is a pollinator that plays important roles in maintaining natural ecosystems and facilitating agricultural production [10]. Over the past 20 years, facility agriculture has rapidly developed along with global agricultural initiatives. Some bumblebee species have been successfully domesticated and raised on a large scale to meet the pollination needs of facility agriculture. As one of the most widely used commercial pollinators globally, *B. terrestris* is economically beneficial to the facility production of fruits and vegetables [11]. Low temperatures induce reproductive diapause in adult *B. terrestris* [12]. In the wild, *B. terrestris* senses temperature changes to terminate diapause and start to development when spring comes. Factory production also mimics natural conditions. Diapause is an important factor restricting colony formation in the artificial mass rearing of *B. terrestris*. The elucidation of the regulatory mechanisms underlying reproductive diapause is a long-standing goal in organismal biology and will facilitate sustainable year-round mass breeding of key pollinator species.

Multiple regulatory mechanisms have been implicated in diapause. Protein phosphorylation is a key regulatory event that is reported to play very important roles in insect physiology, reproduction, and development. In the fruit fly (*Drosophila melanogaster*), calcineurin, a ubiquitous serine/threonine protein phosphatase, broadly influences protein phosphorylation during oocyte maturation and egg activation [7]. Phosphorylation regulatory mechanisms are also involved in the acquisition of thermal tolerance [13]. To date, 27 potential circadian kinases and 789 phosphorylation sites with circadian oscillations have been identified in *D. melanogaster* [14]. In the silkworm (*Bombyx mori*), several key protein-synthesis-related pathways are regulated by phosphorylation, thereby influencing silk production [10]. In the honeybee (*Apis mellifera*), phosphoproteomic analysis has been widely applied to assess the transition from nurse to forager bee, the embryo–larva transition, the production of royal jelly and venom, and the development of the salivary glands and hypopharyngeal glands [15,16,17,18,19,20]. Previous studies have demonstrated that phosphorylation modifications also participate in the regulation of insect diapause [21,22,23,24,25]. A prior report showed significant differences in 23 phosphoproteins in the brain of the cotton bollworm (*Helicoverpa armigera*) in the nondiapause- and diapause-destined pupae [26]. In addition, 27 proteins and phosphoproteins involved in cell proliferation, adult development, and aging were identified in the brain of the flesh fly (*Sarcophaga crassipalpis*) during the initiation of pupal diapause [27].

Proteomic and phosphoproteomic analyses are regarded as powerful approaches for large-scale investigations of gene expression patterns at the protein and post-translational levels. Many proteomic studies have been reported on insect development, immune response, metamorphosis, and diapause [28,29,30,31,32]. However, few studies have looked into the post-translational phosphorylation involved in insect diapause, and even fewer have gone beyond the limitations of two-dimensional electrophoresis (2-DE) coupled with mass spectrometry (MS) in order to identify proteins; this has led to a lack of insight into the regulatory pathway involved in diapause [26,27].

Recently, a high-sensitivity and high-resolution proteome technology, known as the tandem mass tags (TMT) labeling strategy paired with MS, has been developed, allowing for the identification of thousands of proteins with increased coverage and accuracy [33]. Here, a TMT-label-based quantitative analysis was applied to identify differentially expressed proteins (DEPs) and phosphoproteins (DEPPs) of *B. terrestris* queens in the diapause (D), postdiapause (PD), and founder postdiapause (FPD) stages.

## 2. Materials and Methods

### 2.1. Bumblebee Sampling

The *B. terrestris* queens used in this research were purchased from Shandong Lubao Technology Co. Ltd. (Jining, China), an organization dedicated to the commercial production of bumblebees. The sampling was performed as described in a previous study (Chen et al., 2021). In brief, the sampling of *B. terrestris* queens was conducted at three different developmental stages: D—mated queens were maintained at 2 °C in 24 h dark and 50–60% humidity for 10 weeks; PD—diapaused queens were treated with CO_2_ for 1 min, which was repeated once after 24 h at room temperature, and then fed a diet of 50% sucrose for 2 days at 28 °C in 24 h dark and 50–60% humidity; FPD—the postdiapause queens were collected until they began to lay eggs at 28 °C in 24 h dark and 50–60% humidity. All samples were frozen in liquid nitrogen and stored at −80 °C until use. Each group had three biological replicates, and each replicate had three queen bees.

### 2.2. Protein Extraction and Digestion

A schematic overview of the experiment is depicted in Figure 1. One gram of each sample was fully ground into powder in a mortar, which was precooled with liquid nitrogen, then transferred to centrifuge tubes, mixed with four volumes of phenol extraction buffer (10 mM dithiothreitol, 1% protease inhibitor, and 1% phosphatase inhibitor), and sonicated. Following the addition of an equal volume of Tris equilibrium pheno, the sample was centrifuged at 5500× *g* at 4 °C for 10 min. Afterward, the supernatant was collected, mixed with five volumes of 0.1 M ammonium acetate/methanol, and allowed to precipitate overnight. The precipitated proteins were washed with methanol and acetone then redissolved in 8 M urea. The bicinchoninic acid assay was used to measure the protein concentration.

Each protein sample (300 μg) was subjected to enzymatic hydrolysis. In brief, 20% trichloroacetic acid was slowly added to equal amounts of protein and urea lysate then precipitated at 4 °C for 2 h. The supernatant was discarded after centrifugation at 4500× *g* for 5 min, and the precipitation was washed twice with acetone that had been precooled, air-dried, and then mixed with 200 mM triethylammonium bicarbonate buffer and ultrasonicated. Then, trypsin was added at a ratio of 1:50 (protease/protein, m/m), and the mixture was enzymolyzed overnight. The next day, 5 mM dithiothreitol was added, and the mixture was reduced at 56 °C for 30 min, followed by the addition of 11 mM iodoacetamide and incubation at room temperature for 15 min in the dark.

### 2.3. TMT Labeling

After hydrolysis with trypsin as depicted in Figure 1, we desalted the peptides with Strata™ X C18 resin (Phenomenex, Torrance, CA, USA), freeze-dried in vacuum, then dissolved in 0.5 M triethylammonium bicarbonate buffer and labeled using a TMT10plexTM Isobaric Label Reagent Set (Thermo Fisher Scientific, Waltham, MA, USA) in accordance with the manufacturer’s instructions. In brief, the labeled reagent was defrosted, dissolved in acetonitrile, mixed with the peptides, and incubated for 2 h at room temperature. The labeled peptides were mixed, desalted, and freeze-dried in vacuum.

### 2.4. High-pH Peptide Fractionation and Enrichment of Phosphopeptides

High-pH reversed-phase liquid chromatography (RPLC) with a ZORBAX 300Extend-C18 column (particle size, 5 μm; inner diameter, 4.6 mm; length, 250 mm; Agilent Technologies, Inc., Santa Clara, CA, USA) was used to fractionate the peptides. Then, the components were separated with acetonitrile (gradient, 8–32% at pH 9.0) over 60 min into 60 fractions, which were combined into nine fractions and freeze-dried in a vacuum chamber for subsequent analysis.

Peptide mixtures were incubated with a suspension of prewashed immobilized metal chelate affinity chromatography (IMAC) microspheres in a loading buffer (50% acetonitrile/6% trifluoroacetic acid) while vibrating. To elute nonspecifically adsorbed peptides, the IMAC microspheres were successively washed with 50% acetonitrile/6% trifluoroacetic acid and 30% acetonitrile/0.1% trifluoroacetic acid. Afterward, an elution buffer containing 10% NH_4_OH was added to elute the enriched phosphopeptides from the IMAC microspheres while vibrating. The supernatant containing the phosphopeptides was collected and lyophilized for LC-MS/MS analysis.

### 2.5. LC-MS/MS Analysis

The peptides from the holoprotein and enriched phosphopeptides were dissolved in solvent A (0.1% formic acid in 2% acetonitrile) and solvent B (0.1% formic acid in 90% acetonitrile) at a constant flow rate of 500 nL/min with an EASY-nLC 1200 UPLC system. The peptides from the holoproteins were separated with linear gradients of 5–22% solvent B for 36 min, 22–32% solvent B for 16 min, 32–80% solvent B for 4 min, and 80% solvent B for 4 min. The phosphopeptides were separated with linear gradients of 4–20% solvent B for 40 min, 20–32% solvent B for 12 min, 32–80% solvent B for 4 min, and 80% solvent B for 4 min. Afterward, the peptides were subjected to a nanospray ionization source followed by tandem MS with a Q ExactiveTM HF-X Hybrid Quadrupole-OrbitrapTM MS system (Thermo Fisher Scientific) coupled to the EASY-nLC 1200 system. The electrospray voltage was set at 2.2 kV. For peptides from the holoproteins, the m/z scan range was 350 to 1600 with a resolution of 120,000 and a second-order mass scan resolution of 30,000. For the enriched phosphopeptides, the m/z scan range was 350 to 1400 with a resolution of 60,000 and a second-order mass scan resolution of 30,000. In data-dependent collection mode, the 20 most massive precursors were chosen for MS/MS fragmentation by higher-energy collisional dissociation at 28% normalized collision energy. The automatic gain control was set at 3E6 to avoid overfilling the ion trap. The maximum inject time was set to 50 ms, and the dynamic exclusion time was set to 15 s.

### 2.6. Database Search

The MS/MS data that were generated were analyzed by Jingjie PTM Biolab (Hangzhou, China) using a MaxQuant (v.1.5.2.8) quantitative proteomics software package (https://maxquant.org/, accessed on 25 December 2020). Tandem mass spectra were searched against the UniProt database (*B. terrestris*, 17,032 sequences; https://www.uniprot.org/, accessed on 29 December 2020) concatenated with the reverse decoy database.

### 2.7. Gene Ontology (GO), Protein Domain, Kyoto Encyclopedia of Genes and Genomes (KEGG) Pathway Annotation, Subcellular Localization, and Mfuzz Analyses

The UniProt Gene Ontology Annotation database (http://www.ebi.ac.uk/GOA, accessed on 13 March 2021) was used to annotate the GO terms assigned to the proteins, which were classified as cellular components, molecular functions, and physiological processes. The protein domains were annotated with InterProScan software (https://www.ebi.ac.uk/interpro/search/sequence/, accessed on 13 March 2021) with reference to the InterPro domain database (https://www.ebi.ac.uk/interpro/, accessed on 13 March 2021) based on the protein sequence alignment method. The KEGG database (https://www.genome.jp/kegg/pathway.html, accessed on 13 March 2021) was used to annotate the protein pathways. The bioinformatics tool WoLF PSORT (https://wolfpsort.hgc.jp/, accessed on 20 March 2021) was used to predict the subcellular locations of the identified proteins. Genes with similar expression patterns were clustered according the R package Mfuzz (https://www.bioconductor.org/packages/release/bioc/html/Mfuzz.html, accessed on 13 April 2021) [34].

### 2.8. Motif Analysis

Motif-X software (http://motif-x.med.harvard.edu/motif-x.html, accessed on 26 September 2021) was used to examine sequence models composed of amino acids modifying the particular 13-mer locations (six amino acids upstream and downstream) in all protein sequences. All of the protein sequences were searched using background database parameters and other default parameters.

### 2.9. Kinase Analysis

GPS 5.0 software (http://gps.biocuckoo.cn/userguide.php, accessed on 26 September 2021) was used to predict kinase substrate regulation based on the theory that main specificity is provided by short linear motifs around phosphorylation sites. The corresponding kinases were identified by comparison with the kinase sequences in the IEKPD2.0 database (http://iekpd.biocuckoo.org, accessed on 26 September 2021). Potential false-positive hits were filtered in reference to protein–protein interactions (PPIs). A “medium” threshold was chosen for use with the GPS 5.0 software.

The regulatory state of a kinase is reflected by changes to the phosphorylation levels of substrate sites. To predict kinase activity, a gene set enrichment analysis was employed, using log-transformed phosphorylation levels (or ratio values) as a rank file and kinase-phosphorylation site regulations structured as a gmt file for each sample or comparable group. Normalized enrichment scores were regarded as kinase activity scores. Each kinase was considered to have positive activity if the predominant change to the substrate was an increase in phosphorylation and vice versa.

Numerous substrates may be controlled by the same kinase, and a phosphorylation site can be regulated by multiple kinases. Based on the complicated regulatory relationships, a kinase–substrate regulatory network was built using kinases that were predicted to have positive or negative activity and significantly differentially phosphorylated sites.

### 2.10. PPI Networks

All DEPs were searched against the STRING database (version 11.0; https://string-db.org/, accessed on 28 September 2021) for PPIs. Only interactions involving the searched dataset were selected, thereby excluding external candidates. STRING defines the metric “confidence score” as interaction confidence. All interactions with confidence scores ≥ 0.7 (high confidence) were retrieved. The PPI networks were visualized with the R package “networkD3” (https://cran.r-project.org/web/packages/networkD3/index.html, accessed on 28 September 2021).

## 3. Results

### 3.1. Quantitative Proteomic and Phosphoproteomic Analyses of B. terrestris Queens in the D, PD, FPD Stages

The 10-plex TMT isobaric labeling quantitative proteomic method with high-pH reversed phase (high-pH-RP) chromatography coupled with LC-MS/MS was used to investigate and compare differences in protein the abundance and phosphorylation status of the three diapause stages of *B. terrestris*. The principal component analyses (PCA) of both omics datasets showed good clustering of the biological replicates as well as clear separation among the three treatments (Figure 2A,B). In total, 34,760 unique peptides from 4655 proteins were identified, with average sequence coverage of 22% (Figure 2C, Appendix A). The phosphopeptide dataset was first normalized based on the total abundance and further normalized to the corresponding protein abundance for each phosphopeptide. Then, 10,072 unique phosphopeptides were identified from 3339 phosphoproteins with 10,600 phosphorylation sites (Figure 2D, Appendix A). In addition, 1927 proteins overlapped between the proteome and phosphoproteome (Figure 2E). Venn diagrams demonstrate that there were more DEPs in FPD vs. PD than PD vs. D (Figure 2F), with opposite results for DEPPs (Figure 2G). Frequency distributions show that the number and magnitude of changes (log2 ratio) were more significant to the phosphoproteome than the proteome (Figure 2H). The distribution patterns of statistical significance (vertical axis) and log2 fold-change (horizontal axis) for all proteins (Appendix A) and phosphoproteins (Appendix A) identified between the different groups are visualized using volcano plots. Collectively, these results confirmed that phosphorylation plays a crucial role in all stages of diapause.

### 3.2. Expression Profile of Proteins during Different Diapause Stages

Most of the peptides were comprised of 7–20 amino acid residues, and the charge numbers were between 2 and 5 (Appendix A), while the molecular weights varied (Appendix A). A heatmap of Pearson’s correlation coefficients indicated that the proteome was highly reproducible (Appendix A). Collectively, these results indicated that the MS data matched the quality control requirements. The relative quantity of the identified proteins was considered up- or downregulated when the ratio was >1.3 or <0.77 (*p ˂* 0.05). The DEPs of the three groups are summarized in Appendix A. To clarify functional correlations, the DEPs were further classified via KEGG pathway, domain, and GO classification analyses (Appendix A).

To elucidate the protein expression patterns during the three stages of diapause, an Mfuzz clustering analysis was performed based on the abundance levels. In order to screen-out proteins with significant changes, the relative expression levels were first converted to log2 values, and those with SD > 0.4 were screened. The Mfuzz expression pattern clustering analysis (clustering ambiguity “m” = 2) divided the remaining 851 proteins into six clusters (Appendix A). The results of the GO/KEGG/domain enrichment analyses of proteins in different clusters are shown in Appendix A. The representative entries in the GO/KEGG/domain enrichment results are displayed on the right side of the corresponding cluster (Figure 3). The expression levels of 153 and 205 proteins in clusters 1 and 2 were upregulated from D to PD, respectively (Figure 3). The KEGG analysis showed that most of the upregulated proteins correlated to pathways of ribosomes, protein digestion and absorption, steroid hormone biosynthesis, and starch and sucrose metabolism (Appendix A). The biological processes of these proteins are mainly associated with the regulation of transcription, ribosome assembly and biogenesis, cytoplasmic translation, and the post-transcriptional regulation of gene expression (Appendix A). The cellular component analysis showed that these proteins were mainly related to ribosomes, nuclei, and chromosomes (Appendix A). These different genes revealed the reactivation of gene expression and protein translation that emerges from a state of developmental arrest, low metabolic rates, and depressed levels of mRNA. Meanwhile, 89 and 123 proteins in clusters 3 and 5 were downregulated from D to PD, respectively (Figure 3). Most of the downregulated proteins were associated with valine/leucine/isoleucine/tryptophan metabolism and degradation, fatty acid degradation, and glycolysis/gluconeogenesis (Appendix A). Collectively, these results indicated that protein synthesis and energy metabolism increased as the *B. terrestris* queens released from diapause. The most significantly up- and downregulated proteins from PD to FPD were grouped into clusters 6 and 4, respectively (Figure 3). The proteins of cluster 6 were mainly related to fat digestion and absorption, the biosynthesis of unsaturated fatty acids, fatty acid elongation, and protein processing in endoplasmic reticulum, and most of these proteins were located in endoplasmic reticulum (Appendix A). Meanwhile, the KEGG analysis showed that the proteins in cluster 4 were mainly associated with oxidative phosphorylation, the thyroid hormone signaling pathway, and retrograde endocannabinoid signaling (Appendix A). The cellular component analysis showed that these proteins are mainly related to mitochondria (Appendix A). Together, these results suggested that enhanced synthesis of unsaturated fatty acid and protein processing in combination with decreased oxidative phosphorylation may contribute to ovarian maturation in *B. terrestris*.

There were 21 (7 upregulated and 14 downregulated) and 16 upregulated proteins with a fold change > 5 (*p* < 0.05) in the PD/D and FPD/PD groups, respectively. The upregulated DEPs are listed in Table 1. The greatest increase in abundance in the PD/D group (11.5-fold) was observed for a receptor protein associated with the development and maintenance of the mammalian nervous system (LOC100648620: leucine-rich repeat and immunoglobulin-like domain-containing nogo receptor-interacting protein 2) [35]. The remining proteins were annotated with putative roles in the cell cycle (LOC100642207: CDK-activating kinase assembly factor MAT1), cell adhesion and proliferation (LOC100647337: adenomatous polyposis coli), serum Pi level regulation (LOC100647822: inorganic phosphate cotransporter), actin cytoskeleton regulation (LOC100650848: disheveled-associated activator of morphogenesis), protein processing (LOC100643542: E3 ubiquitin-protein ligase Rnf220), and immunity (LOC110119488: transmembrane protease serine). The protein with the greatest increase in the FPD/PD group (10.6-fold) was observed for a calcium-activated chloride channel protein (LOC100646632: anoctamin-1) [36]. Certain immune-associated proteins included serine protease inhibitor 3/4 (LOC100652301), p53 protein (LOC100646232), CLIP domain-containing serine protease (LOC100652157), and TNF receptor-associated factor 5 (LOC105665677). Other proteins with increased expression of at least 5-fold included two lipid metabolic proteins (LOC100645800: pancreatic lipase-related protein 2 and LOC100651730: phospholipase A2), two venom proteins (LOC100642484: venom protease and LOC100647178: venom acid phosphatase Acph-1), two glycometabolism-related proteins (LOC100643608 and LOC100644978), and a major royal jelly protein (LOC100648898).

The expression levels of 14 proteins were significantly reduced in the PD/D group (Appendix A). Two of these proteins play structural roles (LOC100642449: titin and LOC100651429: basement membrane-specific heparan sulfate proteoglycan core protein). The remaining proteins were associated with gene expression regulation (LOC100651892: trithorax; LOC100644496: LIM domain-containing protein jub; and LOC100648151: ataxin-2-like protein) and membrane or channel trafficking (LOC100647837 and LOC100650948). Other proteins with expression levels reduced by at least 5-fold included BMP-binding endothelial regulator (LOC100652150), putative serine/threonine-protein kinase samkC (LOC100649678), immunity-related protein CDV3 (LOC100642536), a storage protein (LOC100650745: hexamerin), vitellogenin-like precursor (LOC100643258), and a protein of unknown function (LOC100644429). In addition, 14 cuticle proteins were significantly decreased in the PD or FPD stage (Appendix A).

### 3.3. Functional Analysis of the Identified Phosphoproteins

The length and charge distribution of the identified phosphoproteins met the quality control requirements (Appendix A). A heatmap of the Pearson’s correlation coefficients indicated that the phosphoproteomes were highly reproducible (Appendix A). Among these, 2241 phosphopeptides corresponding to 1218 proteins, 1021 phosphopeptides corresponding to 692 proteins, and 2152 phosphopeptides corresponding to 1262 proteins were upregulated by more than 1.3-fold between PD/D, FPD/PD, and FPD/D, respectively (Appendix A). To elucidate functional correlations, the DEPPs were enriched by the KEGG pathway, domain, and GO classifications analyses, and the results are showed in Appendix A, respectively.

An Mfuzz clustering analysis was also performed to elucidate the abundance patterns of the phosphoproteins. In total, 3572 phosphoprotein sites were screened (Appendix A). The results of the GO/KEGG/domain enrichment analyses of the phosphoproteins among the different classifications are shown in Appendix A. The proteins assigned to clusters 2 and 3 exhibited increased phosphorylation from D to the PD stage and were selectively enriched for several KEGG terms, including signal transduction (e.g., the calcium signaling pathway, the cyclic guanosine monophosphate (cGMP)-protein kinase G (PKG) signaling pathway, and the hedgehog signaling pathway), neural signal transmission (e.g., the glutamatergic synapse, the GABAergic synapse, and the serotonergic synapse), hormone biosynthesis and secretion (e.g., gonadotropin-releasing hormone secretion, insulin secretion, thyroid hormone synthesis, and cortisol synthesis and secretion), and energy-related metabolism (e.g., the tricarboxylic acid (TCA) cycle) (Figure 4, Appendix A). The detailed GO terms revealed that the biological processes of proteins classified to clusters 2 and 3 were mainly associated with chromosome organization and the regulation of transcription by RNA polymerase II, which is consistent with the changes to the proteome that suggested the reactivation of gene expression (Appendix A). In addition, the enrichment of muscle thin filament assembly and the citrate metabolic process indicated the reactivation as the *B. terrestris* queens released from diapause (Appendix A). The proteins assigned to clusters 1, 4, and 5, which exhibited dephosphorylation from D to the PD stage, were enriched in the KEGG term (mitogen-activated protein kinase) MAPK signaling pathway (Figure 4, Appendix A). Proteins with increased phosphorylation levels from PD to FPD were over-represented in cluster 6 and involved in RNA transport, and the biological process was related to the cell cycle (Figure 4, Appendix A). The KEGG analysis of all DEPPs in Appendix A showed that the upregulated phosphorylated proteins from PD to FPD were involved in energy-related metabolism, including glycerolipid metabolism, glyoxylate and dicarboxylate metabolism, and glycan degradation. Meanwhile, the downregulated phosphorylated proteins were associated with hormone synthesis and secretion, muscle contraction, and the calcium signaling pathway.

In total, 86 (19 upregulated and 67 downregulated) and 10 (9 upregulated and 1 downregulated) phosphorylation sites had fold changes >5 in the PD/D and FPD/PD groups, respectively. The specific upregulated phosphorylation sites are shown in Table 2. Of the 19 phosphorylation sites significantly increased by at least 5-fold in the PD/D groups, 5 phosphorylated proteins were annotated with roles in cytoskeleton and muscle proteins (LOC100648462: catenin; LOC100647948: microtubule-actin cross-linking factor 1; LOC100646259: trichohyalin; LOC100647343: myosin heavy chain; and LOC100643650: titin). Three proteins were annotated as chromatin regulators (LOC100652289: RNA polymerase-associated protein LEO1; LOC100644519: nucleolin; and LOC100648929: lysine-specific demethylase lid). Additional proteins with significantly increased phosphorylation levels included two kinases (LOC100646154: tau-tubulin kinase homolog Asator and LOC100645231: serine/threonine-protein kinase MARK2), two transporter proteins (LOC100649391: solute carrier family 25 member 38 and LOC105665975: golgin subfamily A member 4), transcription- and translation-associated proteins (LOC100647181: glycosyltransferase subunit STT3B; LOC100643117: splicing factor 3B subunit 2; LOC100642874: 60S ribosomal protein L5; and LOC100648934: 60S ribosomal protein L15), and a key component of the ubiquitin ligase protein (LOC100644051: DDB1- and CUL4-associated factor 8). The 67 downregulated phosphorylation sites in the PD/D groups are shown in Appendix A. These 67 phosphorylation sites of 62 proteins were annotated with putative roles in signal transduction mechanisms (*n* = 14), the cytoskeleton (*n* = 8), intracellular trafficking and vesicular transport (*n* = 7), RNA processing and modification (*n* = 5), transcription (*n* = 4), carbohydrate transport and metabolism (*n* = 4), post-translational modification, protein turnover, and chaperones (*n* =4) as well as proteins of unknown function and unannotated proteins (*n* = 17) (Appendix A). Of the nine phosphorylation sites significantly increased by at least 5-fold in the FPD/PD groups, four phosphorylated proteins were annotated with roles in neuroendocrine function (LOC100642774: serine/arginine repetitive matrix protein 2; LOC105666848: jerky protein; LOC100650297: synaptic vesicle membrane protein; and LOC100645647: Katnb1_0 protein), two proteins had roles in sugar metabolism (LOC100644377: sugar transporter ERD6-like 8 and LOC100644978: facilitated trehalose transporter Tret1-like), two proteins had roles in ovarian development (LOC100650436: vitellogenin and LOC100645056: TBC1 domain family member 30), and one protein was annotated as heparanase (LOC100646845).

### 3.4. Kinase Analysis

Phosphoproteomics predicted 760 regulatory relationships between 168 protein kinases and 441 phosphorylated sties on 225 proteins (Appendix A).

It is generally believed that the regulatory state of a kinase is reflected by a change in the phosphorylation levels of its substrate sites. Phosphokinase activity was predicted, and the statistics are shown in Appendix A. A predictive analysis of the kinase activity showed that 11 kinases, including LOC100647871 (serine/threonine kinase 11 (STK11/LKB1)), LOC100647145 (CDK7), and LOC100631091 (JNK), were strongly activated in the D stage, while four and seven kinases were activated in the PD and FPD stages, respectively (Figure 5A).

LOC100647871 (STK11/LKB1) was predicted to phosphorylate A0A6P3U9P6 (Forkhead box protein O (FoxO)) at residues Thr 53 and Thr 205, A0A6P3DMX8 (5′ adenosine monophosphate-activated protein kinase (AMPK) subunit γ-2) at residues Ser 639 and Ser 691, A0A6P3DLX9 (AMPK subunit β-1) at residue Ser 55, and A0A6P3UAT7 (AMPK subunit α-2) at residue Thr 175 (Figure 5C). From D to PD, the activity of LOC100647871 (STK11/LKB1) was blocked, and the phosphorylation level was downregulated.

Kinase activity inferred from phosphoproteomic data indicated that CDK1 (LOC100651604), CDK4 (LOC100650977), CDK14 (LOC100652163), and LSm1 (LOC100651745) were upregulated in the FPD vs. PD group (Figure 5B,D). Meanwhile, the cyclic adenosine monophosphate dependent protein kinase catalytic subunits LOC100652247 and LOC100643510 were downregulated in the FPD vs. PD group (Figure 5B,D). Three upregulated CDK kinases were predicted to phosphorylate DNA replication factor Cdt1, histone RNA hairpin-binding protein, dynamin-1-like protein, lamin-C, and origin recognition complex subunit 1, all of which are associated with the cell cycle and cell division (Figure 5D, Appendix A). LSm1 was predicted to phosphorylate proteins involved with RNA metabolism, such as polyadenylate-binding protein 1, enhancer of mRNA-decapping protein 4, and RNA helicase Me31B. Two downregulated kinases were predicted to phosphorylate dual 3’-5´-cyclic-AMP, -GMP phosphodiesterase 11, and FoxO, which are involved in multiple signal transduction pathways (Figure 5D, Appendix A).

### 3.5. Motif Analysis

Protein kinases prefer specific substrates with conserved motifs. An analysis with Motif-X software predicted 21 upregulated and 23 downregulated conserved motifs among all of the identified phosphorylation sites in the PD vs. D group and 12 upregulated and eight downregulated motifs in the FPD vs. PD group (Appendix A).

Among the upregulated phosphorylation motifs in PD vs. D, the enriched motif contained the acidic amino acid sequence Ser-Asp-x-Glu (the serine was the phosphorylated amino acid residue), which had the highest motif score of 32.00 and the highest fold increase of 11.0 (Table 3). Among the downregulated phosphorylation motifs in PD vs. D, the proline-directed sequence Thr-Pro-Pro (the threonine was the phosphorylated amino acid residue) was strikingly decreased, with a motif score of 27.44 and a fold decrease of 38.2. According to the predicted kinase–substrate regulations, the “…TPP…” sequence was predicted to be recognized by CDK7 (LOC100647145) (Table 3). For the upregulated phosphopeptides in FPD vs. PD, the most statistically significant and enriched motif was also Thr-Pro-Pro, with a motif score of 24.20 and a fold increase of 39.9, which was recognized by CDK1 (LOC100651604) (Table 3). The most significantly downregulated motif in FPD vs. PD was the basic amino acid sequence Lys-x-x-Ser-Pro (Table 3). Heatmaps showing the enrichment and depletion of specific amino acids neighboring the serine/threonine phosphosites were also generated (Appendix A).

### 3.6. PPI and Module Analysis of the DEPs and DEPPs

To further clarify the complicated biological processes regulated in the D, PD, and FPD stages, PPI networks were established based on the identified DEPs and DEPPs. PPI networks for the PD vs. D group are shown in Figure 6, and those for the FPD vs. PD and FPD vs. D groups are shown in Appendix A. The use of the MCODE plugin identified four highly interconnected clusters based on a proteomic analysis of the PD vs. D network that were related to ribosomes, spliceosomes, fructose and mannose metabolism, and glycolysis/gluconeogenesis. Three clusters were identified among the phosphorylated proteins, including proteins related to ribosomes/ribosome biogenesis, oxidative phosphorylation, and metabolic pathways. Taken together, the networks inferred from the proteomic and phosphoproteomic analyses suggest disturbances in transcription, translation, and glycometabolism.

## 4. Discussion

The aim of this study was to create a global overview of the mechanisms underlying diapause, which allows organisms to adapt to unfavorable conditions, with the use of tag-based multiplex proteomic/phosphoproteomic quantitative analyses. The results demonstrated that the termination of diapause from the D to PD stages was closely associated with the neuroendocrine system, energy metabolism, steroid hormone biosynthesis, and various signaling transduction pathways. The transition from PD to the FPD stage was characterized by various biosynthesis and catabolism processes. Kinases and interaction networks were also characterized. According to our best knowledge, this is the first large-scale proteome/phosphoproteome analysis of diapause in bumblebees, and it provides a comprehensive overview of a dynamically intertwined protein and phosphoprotein interaction network.

There have been several studies of the transcriptome and proteome of the different diapause stages of *B. terrestris*. Comparative analyses indicated that diapause in bumblebees is associated with nutrient storage, core metabolic pathways, stress resistance, insect hormone biosynthesis, and proteins involved in cuticle maintenance [37,38,39]. In the current study, comparisons of the protein expression profiles during the three stages of diapause were performed using TMT-labeled proteomics to expand the current understanding of changes at the phosphorylation level.

In arthropods, growing evidence indicates the roles of neurotransmitters in the regulation of diapause. For example, glutamate is reportedly involved in triggering diapause memory in the induction phase of the cotton bollworm (*H. armigera*) [40]. Key proteins involved in neuroactive ligand–receptor interactions and the glutamatergic synapse pathway related to glutamate neurotransmission were upregulated in early diapause in the female red spider mite (*Tetranychus urticae*) [41]. The expression levels of glutamate decarboxylase, which plays a role in neurotransmitter synthesis, were increased in the hemolymph of bumblebee queens at 48 h compared to 6 h postdiapause [39]. In this study, many genes associated with neurotransmitter synthesis were regulated at both the protein and phosphorylation levels during different stages of diapause. KEGG and Mfuzz clustering analyses both revealed that genes involved in neuroactive ligand–receptor interactions (map04080) were over-represented among the upregulated proteins in the PD vs. D group. Compared to the PD to D stage phosphorylation level, the glutamatergic synapse (map04724) and retrograde endocannabinoid signaling (map04723) were over-represented among the upregulated phosphorylated proteins. Indeed, in this study, the phosphorylation levels increased for glutamine synthetase, which is critical for effective neurotransmission; two G-protein subunits, guanine nucleotide-binding protein G (Q) subunit alpha and G(I)/G(S)/G(T) subunit beta-1, which are necessary for G-protein coupling to metabotropic glutamate receptors; and one excitatory amino acid transporter, which is the major transport mechanism for extracellular glutamate in the nervous system [42]. In addition, the GABAergic synapse and the serotonergic synapse were enriched in cluster 2 and upregulated from D to the PD stage, as determined by the phosphoproteomic analysis. Proteins with notable changes in the neuroendocrine system are potential candidates for future mechanistic studies of their upstream roles in insects.

MAPK plays an important role in insect growth, development, and immunity. Rapid condensation, which may be related to MAPK, is important for insets to quickly adapt to low temperatures. A previous study reported that p38, a main member of the MAPK family, regulated the rapid onset and termination of temperature-induced diapause in the flesh fly (*S. crassipalpis*) [43]. At 0 °C, p38 is rapidly activated, consistent with the temperature required for rapid condensation. However, at 25 °C, the phosphorylation of p38 is decreased, indicating that p38 is critical for the rapid response of S. crassipalpis to low temperatures [44]. Extracellular signal-regulated kinase (ERK), another member of the MAPK family, is activated by exposure to cold temperatures and regulates the termination of diapause in the silkworm (*B. mori*) [45]. The expression levels of phosphorylated ERK remain high in the prediapause period, then decrease in the diapause period of the false melon beetle (*Atrachya menetriesi*). Phosphorylated ERK decreases as the temperature increases from 7.5 °C to 25 °C [24]. The results of the present study showed that the phosphorylation level of the MAPK signaling pathway of bumblebees decreased when the temperature was increased from 2 °C (D) to 28 °C (PD), consistent with previous studies. The phosphorylation of RAC (LOC100643457) at residues Ser 157 and Thr 362, PKC (LOC100643492) at residue Thr 57, PAK1 (LOC105666875) at residues Ser 62 and Ser 192, PAK3 (LOC100652006) at residues Ser 2 and Ser 256, MAPKKK15 (LOC100644945) at residue Ser 906, MAPKK (LOC100631092) at residues Ser 689, Ser 419, Ser 701, Ser 245, Ser 714, and Thr 754, and RAF (LOC100645437) at residue Ser 338 were significantly decreased from D to the PD phase. Hence, all these sites are worthy of molecular mechanism studies in the future.

Generally, highly active mitochondria and strong oxidative phosphorylation are hallmarks of growing organisms. However, metabolic quiescence leads to a switch to glycolysis and related gluconeogenesis [46,47]. In the present study, the TCA cycle was enriched in the upregulated phosphorylation group from D to the PD stage, while glycolysis/gluconeogenesis was enriched in the downregulated protein group. These results are consistent with those of previous studies and suggest that increased phosphorylation indicates the activation of oxidative phosphorylation and the TCA cycle. From PD to the FPD stage, oxidative phosphorylation and the TCA cycle were over-represented in both the downregulated proteins and phosphoproteins, which suggests decreased energy demand.

In addition, cuticle proteins were also identified among the DEPs and DEPPs. Of the 24 identified proteins, 14 were significantly decreased at the PD or FPD stage at the protein level. Among the 24 cuticle proteins, three proteins were phosphorylated at 15 sites, including 7 that were decreased at the PD or FPD stage. The characteristic expression profiles of cuticle proteins and phosphorylation modification suggest roles in the recovery from diapause. Other changes were observed to the calcium signaling pathway and the cGMP-PKG signaling pathway, which are reported to play important roles in the reproductive diapause regulation of the red spider mite (*T. urticae*), pupal diapause regulation in the onion fly (*Delia antiqua*), and the termination of diapause in the Chinese citrus fly (*Bactrocera minax*) [41,48,49]. The phosphorylation levels of the calcium signaling pathway and the cGMP-PKG signaling pathway were upregulated from D to PD in the present study. Moreover, other molecular pathways associated with vitamin metabolism were reported to regulate the developmental trajectory associated with diapause in the butterfly (*Phalera bucephala L.*), annual killifish (*Austrofundulus limnaeus*), and American mink (*Neovison vison*) [50,51,52]. In the present study, proteins associated with the metabolism of retinol (vitamin A), ascorbate (vitamin C), aldarate, and vitamin B6 were over-represented at the protein level in the PD vs. D up group, both the PD vs. D and FPD vs. PD up groups, and the FPD vs. PD up group, respectively. These results demonstrate the important roles of vitamin metabolism in the development and diapause of *B. terrestris*.

STK11/LKB1, which is among the most interesting kinases identified, was strongly activated in the D stage but was weakly activated in the PD and FPD stages. LKB1 is a tumor suppressor that regulates multiple biological pathways, including energy metabolism, cell cycle control, and cell polarity by the direct phosphorylation of 14 different AMPK family members [53,54,55,56]. AMPK is a multicomponent enzyme complex that acts as a metabolic stress sensor. To maintain the energy balance, AMPK turns off a number of ATP-using processes once it is active. AMPK is activated by the allosteric binding of AMP, which encourages the binding of upstream kinases to increase activity [57]. Therefore, the down-phosphorylation of AMPK from D to the PD stage by LKB1 facilitates the utilization of ATP to provide energy for development. This is the first report of the phosphorylation of FoxO by LKB1. In contrast, the transcription factor FoxO is reportedly involved in regulating the transcription of LKB1 [58,59]. The transcriptional activity of FoxO is tightly regulated by post-translational modifications, including methylation, acetylation, glycosylation, ubiquitination, and phosphorylation [60,61]. Many kinases (i.e., AKT, PKA, DYRK1, and GSK3) reduce the transcriptional activity of FoxO via phosphorylation [62,63,64]. In response to extracellular stimuli (e.g., insulin, growth factors, and cytokines), AKT phosphorylates and inhibits the transcriptional activity of FoxO via export from the nucleus to the cytoplasm. However, other kinases (i.e., AMPK, MST1, and p38) activate the transcriptional activity of FoxO [65,66,67]. AMPK phosphorylates FoxO at residues Ser 22, Ser 383, and Thr 649 [65,68]. The decreased phosphorylation level of FoxO at residues Thr 53 and Thr 205 from D to the PD stage by LKB1 suggested that these two sites could contribute to the transcriptional activity of downstream diapause-related genes. In addition, the down-phosphorylation of AMPK from D to the PD stage also contributed to the decreased transcriptional activity of FoxO, indicating that LKB1 may act as a switch for the diapause transition.

Our group established a platform based on nanocarrier-mediated RNA interference and the CRISPR/Cas9 system to study the molecular mechanisms of genes involved in reproductive diapause. These DEPs, DEPPs, and kinases could be candidates for RNA interference and CRISPR/Cas9 to manage the diapause of *B. terrestris*, such as shortening or prolonging the diapause time according to the market demand in commercial breeding. The KEGG pathway, enriched with DEPs and DEPPs, such as vitamins and insect hormones, could also be applied to *B. terrestris* to test whether they have a regulatory effect on diapause termination and the preoviposition time.

## 5. Conclusions

In summary, TMT-labeled quantitative proteomics and phosphoproteomics were employed to study dynamic changes in *B. terrestris* at three diapause stages. GO, KEGG pathway enrichment, and Mufzz clustering analyses identified various DEPs and DEPPs related to diapause termination and postdiapause development. Important kinases were identified in the D, PD, and FPD stages. The results of this study will lead to a better understanding of the molecular regulation of diapause in *B. terrestris*.

## Figures and Tables

**Figure 1 insects-13-00862-f001:**
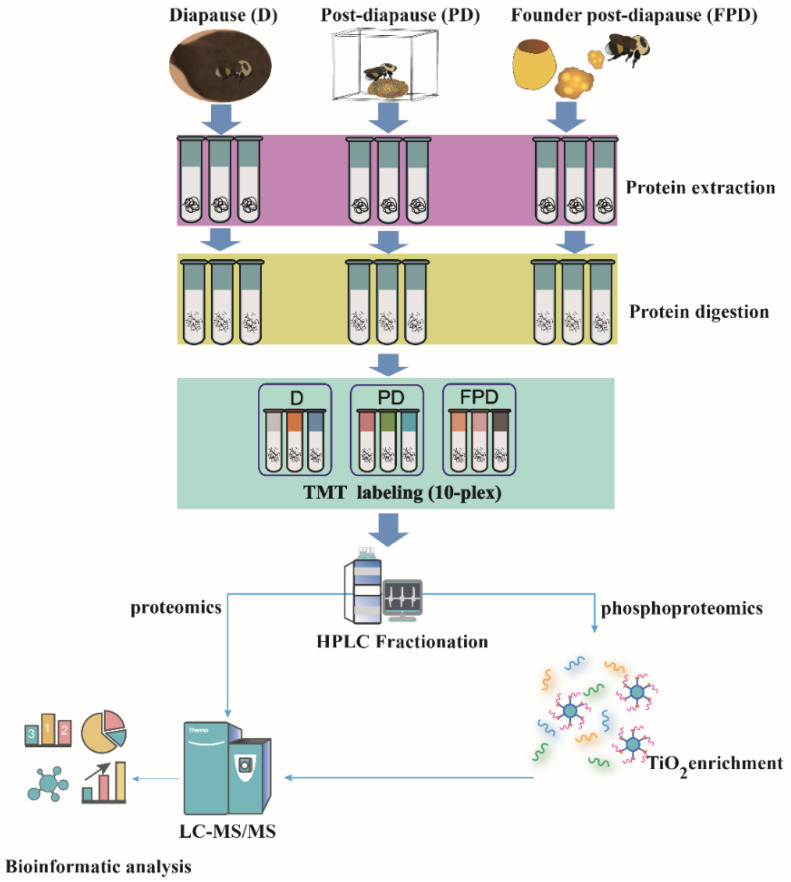
The experimental workflow for the quantitative proteomics and phosphoproteomics analyses of three diapause stages in *B. terrestris*.

**Figure 2 insects-13-00862-f002:**
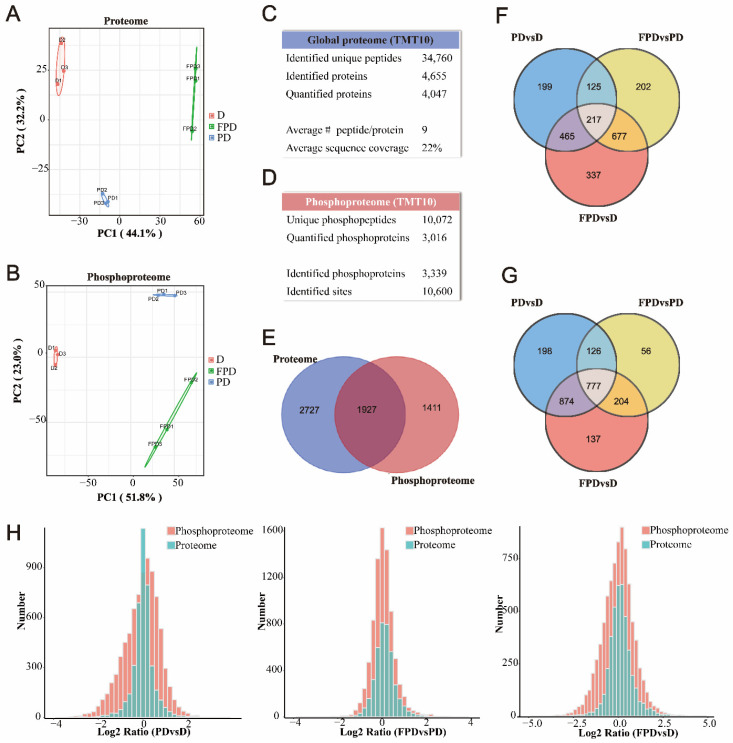
Overview of the proteomes and phosphoproteomes of the three diapause stages in *Bombus terrestris*. (**A**,**B**) Principal component analysis (PCA) of quantified proteins and quantified phosphopeptides from D, PD, and FPD. (**C**,**D**) The summary information of proteome/phosphoproteome. (**E**) Venn diagram of proteins and phosphoproteins identified in proteome and phosphoproteome experiments, respectively. (**F**,**G**) Venn diagram illustrating differentially expressed proteins or phosphoproteins in each group comparison (PD vs. D, FPD vs. PD, and FPD vs. D). (**H**) Distribution histograms of log fold-change comparing proteomes (green) and phosphoproteomes (red) identified in three comparable groups.

**Figure 3 insects-13-00862-f003:**
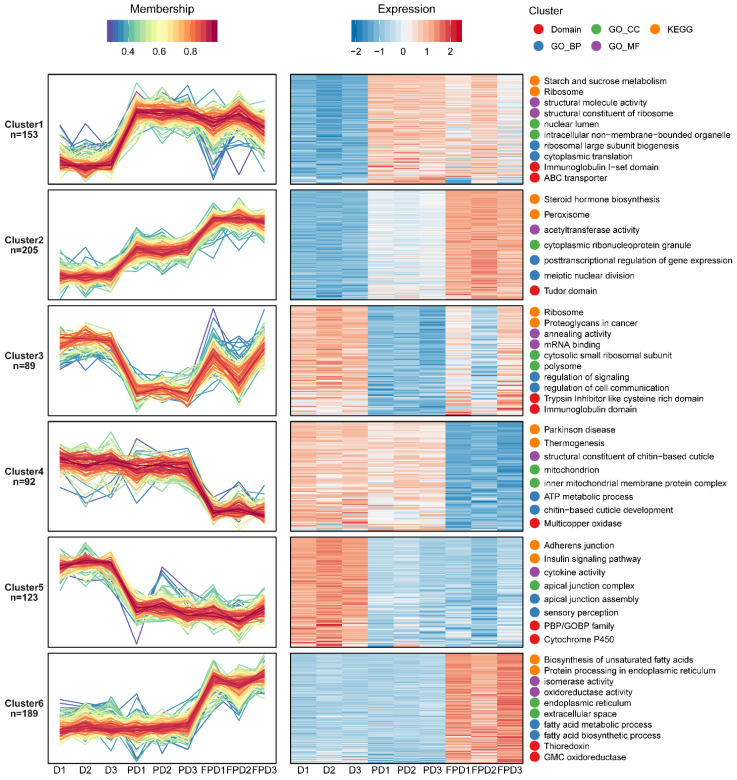
Functional clustering of differentially expressed proteins. Fuzzy clustering algorithm was used to identify coregulated protein modules. The two most highly enriched GO/KEGG/domain entries are displayed on the right side of the corresponding cluster graph.

**Figure 4 insects-13-00862-f004:**
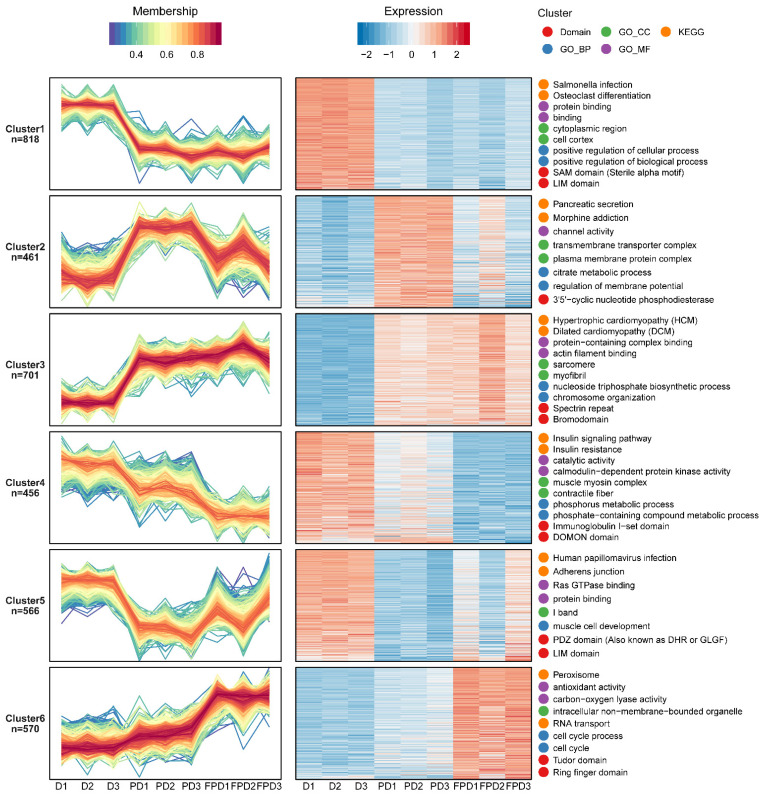
Functional clustering of differentially expressed phosphoproteins. A fuzzy clustering algorithm was used to identify coregulated phosphoprotein modules. The two most highly enriched GO/KEGG/domain entries are displayed on the right side of the corresponding cluster graph.

**Figure 5 insects-13-00862-f005:**
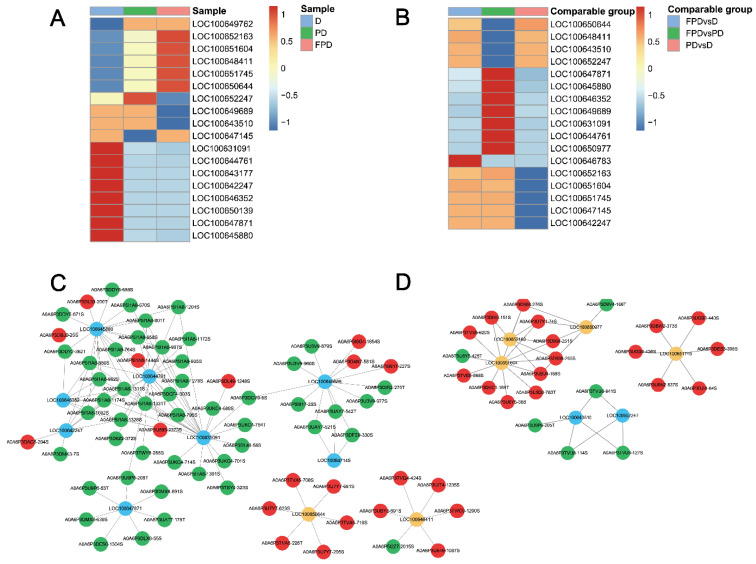
Changes in kinase activity and the kinase–substrate regulatory network in the different diapause stages of queens. (**A**) Kinase activity in D, PD, and FPD. Sample names in columns and phosphokinases in rows (rows are clustered and Z-score-normalized). (**B**) Kinase activity matrix heat map in comparable groups. Phosphokinases are listed in rows, and comparable groups are listed in columns s (rows are clustered and Z-score-normalized). The key kinase–substrate regulatory networks in (**C**) PD/D and (**D**) FPD/PD. Orange represents activated kinases, blue represents inhibited kinases, red represents upregulated phosphorylation sites, and green represents downregulated phosphorylation sites.

**Figure 6 insects-13-00862-f006:**
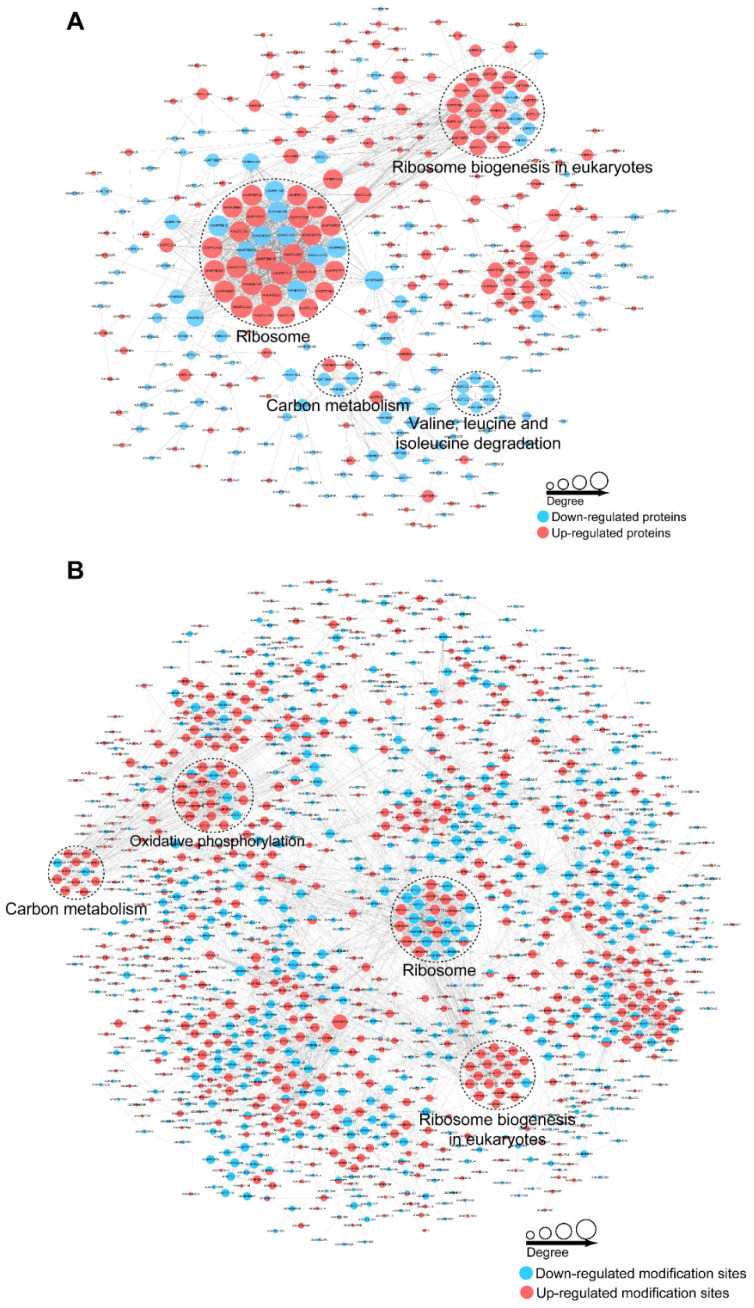
Protein–protein interaction networks of DEPs (**A**) and DEPPs (**B**) between PD/D.

**Table 1 insects-13-00862-t001:** The upregulated DEPs in PD/D and FPD/PD with fold changes higher than 5.

Protein Accession	Protein Description	Ratio	*p* Value	Gene Name	MW (kDa)
Comparable group PD/D up					
A0A6P3D6E6	leucine-rich repeat and immunoglobulin-like domain-containing nogo receptor-interacting protein 2	11.493	5.98 × 10^−5^	LOC100648620	103.4
A0A6P3DB00	CDK-activating kinase assembly factor MAT1	8.873	1.055 × 10^−5^	LOC100642207	36.711
A0A6P5HS32	Transmembrane protease serine	5.623	0.0093336	LOC110119488	65.551
A0A6P3UB32	putative inorganic phosphate cotransporter	5.601	9.643 × 10^−5^	LOC100647822	53.886
A0A6P3TTL0	suppressor APC domain-containing protein 2	5.545	0.0015442	LOC100647337	51.622
A0A6P5I390	E3 ubiquitin-protein ligase Rnf220	5.301	0.0002411	LOC100643542	35.027
A0A6P3DC42	disheveled-associated activator of morphogenesis 1	5.122	9.54 × 10^−6^	LOC100650848	123.68
Comparable group FPD/PD up					
A0A6P5IBX2	anoctamin-1	0.551	0.0077095	LOC100646632	115.18
A0A6P3DGE6	pancreatic lipase-related protein 2	0.99	0.8897713	LOC100645800	34.752
A0A6P3U8X6	phospholipase A2	1.159	0.2525801	LOC100651730	22.336
A0A6P3TVD7	cation-independent mannose-6-phosphate receptor	0.898	0.098125	LOC100651480	82.845
A0A6P3DNC4	serine protease inhibitor 3/4	1.356	0.0453889	LOC100652301	35.236
A0A6P3DFC5	nuclear factor related to kappa-B-binding protein	1.138	0.1405269	LOC100649343	168.64
A0A6P5HLS9	protein toll	1.775	0.0262693	LOC100651716	125.92
A0A6P3U2L5	Apoptosis-stimulating of p53	1.607	0.02605	LOC100646232	139.27
A0A6P3TPE5	venom protease	1.551	0.0013554	LOC100642484	32.966
A0A6P3TWR2	TNF receptor-associated factor 5	2.023	0.0232261	LOC105665677	55.742
A0A6P3UKF1	venom acid phosphatase Acph-1	1.208	0.1376568	LOC100647178	48.343
A0A6P5I2Z1	major royal jelly protein 1	2.115	0.0755501	LOC100648898	41.083
A0A6P3UGV4	basement membrane-specific heparan sulfate proteoglycan core protein	0.067	0.0001312	LOC100651429	468.9
A0A6P5HLF4	CLIP domain-containing serine protease	1.181	0.0913721	LOC100652157	82.898
A0A6P3TNQ5	alpha-glucosidase	1.321	0.0721932	LOC100643608	65.637
A0A6P3DF18	facilitated trehalose transporter Tret1-lik	0.811	0.1431761	LOC100644978	76.971

**Table 2 insects-13-00862-t002:** The upregulated DEPPs in PD/D and FPD/PD with fold changes higher than 5.

Protein Accession	Position	Amino Acid	Protein Description	Ratio	*p* Value	Gene Name
Comparable group PD/D up						
A0A6P3U5V0	91	S	RNA polymerase-associated protein LEO1	9.982	4.011 × 10^−5^	LOC100652289
A0A6P3DA37	471	S	dolichyl-diphosphooligosaccharide--protein glycosyltransferase subunit STT3B	9.306	0.0002187	LOC100647181
A0A6P5IE28	166	S	catenin alpha	8.092	0.0001623	LOC100648462
A0A6P5I410	505	T	splicing factor 3B subunit 2	8.01	0.0005079	LOC100643117
A0A6P3TWI1	591	S	DDB1- and CUL4-associated factor 8	7.232	0.0001319	LOC100644051
A0A6P3U5V0	119	S	RNA polymerase-associated protein LEO1	6.862	0.0001676	LOC100652289
A0A6P3DB01	580	S	nucleolin	6.187	3.099 × 10^−6^	LOC100644519
A0A6P3UJ38	26	S	GD14428	6.181	3.796 × 10^−5^	LOC100643604
A0A6P3UFI4	228	T	microtubule-actin cross-linking factor 1	5.57	6.545 × 10^−5^	LOC100647948
A0A6P3UAY9	147	S	solute carrier family 25 member 38	5.54	4.793 × 10^−5^	LOC100649391
A0A6P3UJ22	273	S	60S ribosomal protein L5	5.503	0.006308	LOC100642874
A0A6P5HLM2	926	S	trichohyalin	5.485	9.834 × 10^−5^	LOC100646259
A0A6P3U0R3	2381	S	golgin subfamily A member 4	5.469	0.0001002	LOC105665975
A0A6P5I2T4	771	S	tau-tubulin kinase homolog Asator	5.344	0.0068525	LOC100646154
A0A6P5HGT7	692	S	serine/threonine-protein kinase MARK2 isoform X11	5.248	0.000345	LOC100645231
A0A6P5HR14	1421	T	myosin heavy chain, muscle	5.237	0.0001658	LOC100647343
A0A6P3D774	1536	T	lysine-specific demethylase lid	5.103	0.0060939	LOC100648929
A0A6P5HR09	5303	T	titin	5.094	0.0011792	LOC100643650
A0A6P3D7R3	197	S	60S ribosomal protein L15	5.046	0.0008772	LOC100648934
Comparable group FPD/PD up						
A0A6P3TLJ9	504	T	vitellogenin	10.046	0.0048077	LOC100650436
A0A6P3U9T5	23	S	sugar transporter ERD6-like 8	8.168	0.0001362	LOC100644377
A0A6P3U996	454	S	serine/arginine repetitive matrix protein 2	7.991	0.0094707	LOC100642774
A0A6P3UGK1	319	S	jerky protein homolog-like	7.445	0.0183674	LOC105666848
A0A6P3U6P7	1604	S	TBC1 domain family member 30	5.581	0.1101188	LOC100645056
A0A6P3DHB6	22	T	synaptic vesicle membrane protein VAT-1 homolog-like	5.525	0.0003307	LOC100650297
A0A6P3DFL1	81	S	Katnb1_0 protein	5.434	0.0002241	LOC100645647
A0A6P3DF18	681	S	facilitated trehalose transporter Tret1-like	5.329	0.0006047	LOC100644978
A0A6P3UBL9	551	S	Heparanase	5.221	0.0251118	LOC100646845

**Table 3 insects-13-00862-t003:** The representative phosphorylation motifs on differentially expressed phosphopeptide sequences.

Group	Motif Logo	Motif	Motif Score	Fold Change	Putative Protein Kinases
PD vs. D up	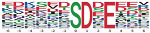	Xxxxxx_S_DxExxx	32.00	11.0	novel
PD vs. D down	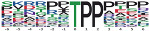	Xxxxxx_T_PPxxxx	27.44	38.2	cyclin dependent kinase 7 substrate motif
FPD vs. PD up	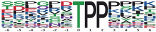	Xxxxxx_T_PPxxxx	24.2	39.9	cyclin dependent kinase 1
FPD vs. PD down	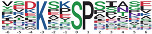	xxxKxx_S_Pxxxxx	25.46	14.8	novel

## Data Availability

The mass spectrometry proteomics and phosphoproteomics data were deposited to the ProteomeXchange Consortium (http://proteomecentral.proteomexchange.org, accessed on 26 May 2022) via the iProX partner repository [69] with the dataset identifier PXD034198 (proteome) and PXD034199 (phosphoproteome).

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
