# Peer review of "Integrative Proteomic and Phosphoproteomic Analyses Revealed Complex Mechanisms Underlying Reproductive Diapause in Bombus terrestris Queens"

_insects, 2022, doi:10.3390/insects13100862_

Round 1

Reviewer 1 Report

2. Materials and Methods

In section 2.1 

The author should mentioned the photoperiod and Relative humidity during the induction of diapause in Bumblebee. It implies also in post diapuase as well as in reproductive diapause.

Author Response

Dear reviewer,

Thank you for your positive attitude and the great comments! We are resubmitting a revised version of our manuscript " Integrative proteomic and phosphoproteomic analyses revealed complex mechanisms underlying reproductive diapause in Bombus terrestris queens" (insects-1876268).

As you required, we further revised the manuscript to address the concern. All the changes in the revised version were made using "Track changes". We carefully answered all questions and believe that the revised manuscript has been substantially improved.

The following is the point-by-point response to the reviewer’s comments.

Best regards,

Yifan Zhai

Institute of Plant Protection

Shandong Academy of Agricultural Sciences

Jinan 250100

China

Tel: +86-0531-66655315, Email: [email protected].

Response to Reviewer Comments

Point 1: In section 2.1. The author should mentioned the photoperiod and Relative humidity during the induction of diapause in Bumblebee. It implies also in post diapuase as well as in reproductive diapause.

Response 1:Thank you for your kind comment! The photoperiod is darkness all day and the humidity is 50%-60% in D, PD and FPD. I have revised the manuscript at Line 127-131 of Page 4.

Reviewer 2 Report

Yang Liu's paper addressed an important aspect in the understanding of some biological phenomena including diapause. 

My main concern is what do the authors make as a link between the proteins found and diapause? has climate change changed this phenomenon for example? or from the proteins detected, what do the authors propose in the management of diapause of this strong pollinating insect?

Others Remarks

Ligne 23 : TMT labeled. What does TMT mean?

write all scientific names: Apis mellifera, Bombyx mori, D. melanogaster....meme B. terrestris in italics.

Ligne 135: Equal amounts of protein of each sample. How much amounts?

Ligne 248: schematic overview of the experiment is depicted 248 in Figure 1.: this figure should be in the materials and methods section.

Table 2. The up-regulated DEPPs in PD/D and FPD/PD with a fold change higher than 5.: Is there a possibility to add the biological functions?

In the discussion, it is not necessary to quote the sources of the results (figure 1, table 3.........), it is enough to announce the results and compare them with other studies.

Author Response

Dear reviewer,

Thank you for your positive attitude and the great comments! We are resubmitting a revised version of our manuscript " Integrative proteomic and phosphoproteomic analyses revealed complex mechanisms underlying reproductive diapause in Bombus terrestris queens" (insects-1876268).

As you required, we further revised the manuscript. All the changes in the revised version were made using "Track changes". We carefully answered all questions and believe that the revised manuscript has been substantially improved.

The following is the point-by-point response to the reviewer’s comments.

Best regards,

Yifan Zhai

Institute of Plant Protection

Shandong Academy of Agricultural Sciences

Jinan 250100

China

Tel: +86-0531-66655315, Email: [email protected].

Response to Reviewer Comments

Point 1:My main concern is what do the authors make as a link between the proteins found and diapause? has climate change changed this phenomenon for example? or from the proteins detected, what do the authors propose in the management of diapause of this strong pollinating insect?

Response 1:Thank you for your concern! Diapause is a dynamic process consisting of several phases which could be divided into pre-diapause, diapause and post-diapause. Pre-diapause could be subdivided into induction phase and preparation phase. Diapause could be subdivided into initiation, maintenance and termination. Post-diapause is followed by the termination of diapause. The DEPs and DEPPs found in this paper would contribute to diapause termination and post-diapause development and reproduction regulation. In the wild, Bombus terrestris sense the temperature changes to terminate diapause and start to development when spring comes. Factory production also mimics natural conditions, so the climate change will change the phenomenon in the wild.

From the proteins detected, all DEPs could be candidates for RNA interference and CRISPR/Cas9 to manage the diapause of B. terrestris, such as shorten or prolong diapause time according to the market demand in commercial breeding. The KEGG pathway enriched with DEPs and DEPPs, such as vitamins and insect hormones, could also be applied to B. terrestris to test whether they have a regulatory effect on diapause termination and preoviposition time. These are what we want to do in the future.

Ponit 2: Line 23 : TMT labeled. What does TMT mean?

Response 2: The full name of TMT is Tandem Mass Tags, which is a novel MS/MS-based analysis strategy using isotopomer labels for the accurate quantification of peptides and proteins. All TMT labels have the same chemical structure and are composed of reporter, equilibrium, and peptide-reactive groups, but each label contains isotopes substituted at different positions, so the reporter and equilibrium groups have different molecular masses in each label. The combined groups then have the same total molecular weight and structure, so that molecules labeled with different labels are not distinguishable during chromatographic or electrophoretic separation, as well as in a single MS mode. In MS/MS mode, the sequence information can be obtained from the fragment ions, and the quantitative data can be obtained from the fragment of the tag, so as to realize the quantitative analysis of the proteome.

Ponit 3: write all scientific names: Apis mellifera, Bombyx mori, D. melanogaster....meme B. terrestris in italics.

Response 3: Thank you for your comment! I have revised all the scientific names in italics.

Ponit 4: Ligne 135: Equal amounts of protein of each sample. How much amounts?

Response 4: Thank you for your comment! The amounts is 300 μg. I have supplied the information in the revised manuscript (Page 4, Line 146).

Ponit 5: Ligne 248: schematic overview of the experiment is depicted 248 in Figure 1.: this figure should be in the materials and methods section.

Response 5: Thank you for your advice! I have move it to the materials and methods section (Page 4, Line 135).

Ponit 6: Table 2. The up-regulated DEPPs in PD/D and FPD/PD with a fold change higher than 5.: Is there a possibility to add the biological functions?

Response 6: Thanks for your advice. I have add the biological proecss and molecular functions of up-regulated DEPs and DEPPs in the supplementary table 6 and 8, respectively.

Ponit 7: In the discussion, it is not necessary to quote the sources of the results (figure 1, table 3.........), it is enough to announce the results and compare them with other studies.

Response 7: Thanks for your kind advice! As you said, most of the articles do not need to quote the sources in the discussion, but considering that there are 14 supplementary figures and 12 supplementary tables in this article, the description of the details in the main text are limited, the quotation is added in the discussion for the convenience of finding the results.

Reviewer 3 Report

The authors utilized the TMT-labeled quantitative proteomics and phosphoproteomics to identify various DEPs and DEPPs related to the diapause termination and post-diapause development the bumblebee, Bombus terrestris, one of the most ideal pollinators in the world. Through this research, a set of important kinases were also identified in the D, PD, and FPD stages. The results of this study could lead to a better understanding of the molecular regulation of diapause in B. terrestris. In general, the present data is convincing and the manuscript is in good shape in term of writing. It will certainly improve the quality of the manuscript if the authors could provide a case study to verify a differentially expressed protein or phosphoprotein such as a kinase identified in proteomics and phosphoproteomics analyses using Western blot or a similar approach.

Specific comments (mainly editorial)

Abstract

1.       L29: “a combination analyses” to “a combination analysis”.

Introduction

2.       L81: “Drosophila melanogaster” should be italicized. The authors italicized Bombus terrestris, but not the rest of species. Please italicize them accordingly throughout the manuscript.

3.       L83: “Phosphorylation regulatory mechanisms also involved in the acquisition of” to “Phosphorylation regulatory mechanisms is also involved in the acquisition of” or “Phosphorylation regulatory mechanisms also involve the acquisition of”.

4.       L104: “which resulting in” to “resulting in” or “which results in”.

Materials and Methods

5.       L116: “B.terrestris”. insert a space in between.

6.       L120: “CO2” to “CO2”.

7.       L126: “An appropriate amount of each sample”? please be more specific about the quantity of each sample.

8.       L133: “redissolved with” to “redissolved in”.

9.       L136: “protein and lysate”. What is lysate? No explanation of lysate prior to this point.

10.   L149: “In brief, the labeled reagent was defrosted and dissolved in acetonitrile, then mixed with the peptides and incubated at room temperature for 2 h” to “In brief, the labeled reagent was defrosted, dissolved in acetonitrile, mixed with the peptides, and then incubated at room temperature for 2 h”.

11.   L155: “then separated” to “and then separated”.

12.   L164: “NH4OH” to “NH4OH”.

13.   L208: “sequences models” to “sequence models”.

14.   L222: “was formatted” to “were formatted”.

15.   L228: “significantly differentially expressed phosphorylation sites” to “significantly differentially phosphorylated sites”.

16.   L240-242: It is suggested to delete these sentences.

Results

17. L287 and 334: “(p<0.05)”. Please italicize p i.e. (p<0.05).

18. L307: “chromosome” to “and chromosome”.

19. L323: “these proteins mainly related to” to “these proteins are mainly related to”.

20. L335: “were listed” to “are listed”.

21. L345: “(10.6-fold)was observed”. Insert a space in front of was.

22. L442: “Tret1-like), two proteins” to “Tret1-like), and two proteins”.

23. L451: “by change to the phosphorylation levels” to “: “by change at the phosphorylation levels”.

24. L452: “The greater the phosphorylation level of a substrate site, the greater the phosphokinase activity, and vice versa”. This statement may not be always true. Protein phosphatase is also influencing the phosphorylation levels. What you observed is the balance of protein kinase activity and protein phosphatase activity.  

Discussion

25. L562: “two G-protein subunit” to “two G-protein subunits”.

Author Response

Dear reviewer,

Best regards

Round 2

Reviewer 1 Report

The manuscript is well presented.

Author Response

Dear reviewer,

  Thank you for your positive attitude of the manuscript.

Best regards,

Yifan Zhai

Reviewer 2 Report

Point 1:My main concern is what do the authors make as a link between the proteins found and diapause? has climate change changed this phenomenon for example? or from the proteins detected, what do the authors propose in the management of diapause of this strong pollinating insect?

Response 1:Thank you for your concern! Diapause is a dynamic process consisting of several phases which could be divided into pre-diapause, diapause and post-diapause. Pre-diapause could be subdivided into induction phase and preparation phase. Diapause could be subdivided into initiation, maintenance and termination. Post-diapause is followed by the termination of diapause. The DEPs and DEPPs found in this paper would contribute to diapause termination and post-diapause development and reproduction regulation. In the wild, Bombus terrestris sense the temperature changes to terminate diapause and start to development when spring comes. Factory production also mimics natural conditions, so the climate change will change the phenomenon in the wild.

From the proteins detected, all DEPs could be candidates for RNA interference and CRISPR/Cas9 to manage the diapause of B. terrestris, such as shorten or prolong diapause time according to the market demand in commercial breeding. The KEGG pathway enriched with DEPs and DEPPs, such as vitamins and insect hormones, could also be applied to B. terrestris to test whether they have a regulatory effect on diapause termination and preoviposition time. These are what we want to do in the future.

I don't see your comment in the text. The suggestion is to enrich the manuscript, not just for me

Ponit 2: Line 23 : TMT labeled. What does TMT mean?

Response 2: The full name of TMT is Tandem Mass Tags, which is a novel MS/MS-based analysis strategy using isotopomer labels for the accurate quantification of peptides and proteins. All TMT labels have the same chemical structure and are composed of reporter, equilibrium, and peptide-reactive groups, but each label contains isotopes substituted at different positions, so the reporter and equilibrium groups have different molecular masses in each label. The combined groups then have the same total molecular weight and structure, so that molecules labeled with different labels are not distinguishable during chromatographic or electrophoretic separation, as well as in a single MS mode. In MS/MS mode, the sequence information can be obtained from the fragment ions, and the quantitative data can be obtained from the fragment of the tag, so as to realize the quantitative analysis of the proteome.

I meant when the abbreviation of the word appears for the first time, it must be written in full

Ponit 5: Ligne 248: schematic overview of the experiment is depicted 248 in Figure 1.: this figure should be in the materials and methods section.

Response 5: Thank you for your advice! I have move it to the materials and methods section (Page 4, Line 135).

The figure is always in the result part of your V2 document.

You should not lie, to say that you have taken into account when it is not the case.

Ponit 7: In the discussion, it is not necessary to quote the sources of the results (figure 1, table 3.........), it is enough to announce the results and compare them with other studies.

Response 7: Thanks for your kind advice! As you said, most of the articles do not need to quote the sources in the discussion, but considering that there are 14 supplementary figures and 12 supplementary tables in this article, the description of the details in the main text are limited, the quotation is added in the discussion for the convenience of finding the results.

I don't agree with the authors at all. 

The references of figures and tables are quoted in the RESULT section and not in the discussion. If the authors want to do this, the editors should not send me this manuscript anymore

Author Response

Dear reviewer,

Thank you for all of your comments! We are resubmitting a revised version of our manuscript " Integrative proteomic and phosphoproteomic analyses revealed complex mechanisms underlying reproductive diapause in Bombus terrestris queens" (insects-1876268).

According to your suggestions, we further revised the manuscript. All the changes in the revised version were made using "Track changes". We carefully answered all questions and believe that the revised manuscript has been substantially improved.

The following is the point-by-point response to the reviewer’s comments in blue font.

Best regards,

Yifan Zhai

Institute of Plant Protection

Shandong Academy of Agricultural Sciences

Jinan 250100

China

Tel: +86-0531-66655315, Email: [email protected].

Response to Reviewer Comments

Point 1:My main concern is what do the authors make as a link between the proteins found and diapause? has climate change changed this phenomenon for example? or from the proteins detected, what do the authors propose in the management of diapause of this strong pollinating insect?

Response 1:Thank you for your concern! Diapause is a dynamic process consisting of several phases which could be divided into pre-diapause, diapause and post-diapause. Pre-diapause could be subdivided into induction phase and preparation phase. Diapause could be subdivided into initiation, maintenance and termination. Post-diapause is followed by the termination of diapause. The DEPs and DEPPs found in this paper would contribute to diapause termination and post-diapause development and reproduction regulation. In the wild, Bombus terrestris sense the temperature changes to terminate diapause and start to development when spring comes. Factory production also mimics natural conditions, so the climate change will change the phenomenon in the wild.

From the proteins detected, all DEPs could be candidates for RNA interference and CRISPR/Cas9 to manage the diapause of B. terrestris, such as shorten or prolong diapause time according to the market demand in commercial breeding. The KEGG pathway enriched with DEPs and DEPPs, such as vitamins and insect hormones, could also be applied to B. terrestris to test whether they have a regulatory effect on diapause termination and preoviposition time. These are what we want to do in the future.

I don't see your comment in the text. The suggestion is to enrich the manuscript, not just for me.

Response: Thank you for your suggestion! The sentence of “Diapause is a dynamic process consisting of several phases which could be divided into pre-diapause, diapause and post-diapause. Pre-diapause could be subdivided into induction phase and preparation phase. Diapause could be subdivided into initiation, maintenance and termination. Post-diapause is followed by the termination of diapause” has been included in the manuscript (Page 2, Line 55-58). The sentence of “In the wild, Bombus terrestris sense the temperature changes to terminate diapause and start to development when spring comes. Factory production also mimics natural conditions” has been inserted into Page 2, Line 75-77. The sentence of “all DEPs could be candidates for RNA interference and CRISPR/Cas9 to manage the diapause of B. terrestris, such as shorten or prolong diapause time according to the market demand in commercial breeding. The KEGG pathway enriched with DEPs and DEPPs, such as vitamins and insect hormones, could also be applied to B. terrestris to test whether they have a regulatory effect on diapause termination and preoviposition time” has been inserted into Page 23, Line 692-697.

Ponit 2: Line 23 : TMT labeled. What does TMT mean?

Response 2: The full name of TMT is Tandem Mass Tags, which is a novel MS/MS-based analysis strategy using isotopomer labels for the accurate quantification of peptides and proteins. All TMT labels have the same chemical structure and are composed of reporter, equilibrium, and peptide-reactive groups, but each label contains isotopes substituted at different positions, so the reporter and equilibrium groups have different molecular masses in each label. The combined groups then have the same total molecular weight and structure, so that molecules labeled with different labels are not distinguishable during chromatographic or electrophoretic separation, as well as in a single MS mode. In MS/MS mode, the sequence information can be obtained from the fragment ions, and the quantitative data can be obtained from the fragment of the tag, so as to realize the quantitative analysis of the proteome.

I meant when the abbreviation of the word appears for the first time, it must be written in full.

Response: I’m sorry for the misunderstanding. I have corrected in the revised manuscript (Page 1, Line 24).

Ponit 5: Ligne 248: schematic overview of the experiment is depicted 248 in Figure 1.: this figure should be in the materials and methods section.

Response 5: Thank you for your advice! I have move it to the materials and methods section (Page 4, Line 135).

The figure is always in the result part of your V2 document.

You should not lie, to say that you have taken into account when it is not the case.

Response: Thank you for your comment! I’m sorry that I did not explain clearly. I mean that I have moved the quotation to the materials and methods section. I didn’t move the figure because I thought that the figure will be rearranged by editor according to the layout before published. And I have moved the figure to the materials and methods section now.

Ponit 7: In the discussion, it is not necessary to quote the sources of the results (figure 1, table 3.........), it is enough to announce the results and compare them with other studies.

Response 7: Thanks for your kind advice! As you said, most of the articles do not need to quote the sources in the discussion, but considering that there are 14 supplementary figures and 12 supplementary tables in this article, the description of the details in the main text are limited, the quotation is added in the discussion for the convenience of finding the results.

I don't agree with the authors at all. 

The references of figures and tables are quoted in the RESULT section and not in the discussion. If the authors want to do this, the editors should not send me this manuscript anymore

Response: Thank you for your comment! I have deleted the quotations in the discussion.

Reviewer 3 Report

The authors properly addressed all concerns raised by this reviewer and I have no further major concerns other than a couple very minor editorial errors as listed below:

L128: “1 g of” to “One gram of”. Do not use numeric number at the beginning of a sentence.

L 138: “300 μgprotein of each sample” to “Each protein sample (300 μg)” to avoid using numeric number at the beginning of a sentence.

Since these issues are minor, they can be addressed  during the proofreading process. 

Author Response

Dear reviewer,

  Thank you for your positive attitude of the manuscript, and I have corrected the errors in the revised manuscript.

Best regards,

Yifan Zhai